# Design and Application of a Twisted and Coiled Polymer Driven Artificial Musculoskeletal Actuation Module

**DOI:** 10.3390/ma15228261

**Published:** 2022-11-21

**Authors:** Chunbing Wu, Wen Zheng, Zhiyi Wang, Biao Yan, Jia Ma, Guangqiang Fang

**Affiliations:** 1Space Structure and Mechanism Technology Laboratory of China Aerospace Science and Technology Group Co., Ltd., Institute of Aerospace System Engineering Shanghai, Shanghai 201108, China; 2Institute of Biomedical Manufacturing and Life Quality Engineering, School of Mechanical Engineering, Shanghai Jiao Tong University, Shanghai 200240, China

**Keywords:** artificial muscle, soft actuator, twisted and coiled polymer, soft robot

## Abstract

Twisted and coiled polymer (TCP) artificial muscles can exhibit unidirectional actuation similar to skeletal muscles. This paper presents a TCP driven artificial musculoskeletal actuation module that can be used in soft robots. This module can contract in the axis direction, and the contraction displacement and force can be controlled easily. The main body of the actuation module consists of TCP muscles and leaf springs, and the deformation of the module is actuated by the TCP muscles. A prototype was made to test the performance of the module. The design and experimental results of the module are presented. The module can provide contraction motion. Results show that the module can provide a contraction force of 0.7 N with displacement of approximately 6.8 mm at 120 °C when exposed to electrical power of 24 V. The proposed artificial musculoskeletal actuation module can potentially be applied in biomimetic robots and the aerospace field.

## 1. Introduction

Skeletal muscles are key parts of the natural biology system and have remarkable and versatile maneuverability [1]. They can contract and pull the skeleton to complete various types of body motions. Inspired by such natural muscles, researchers have developed various soft actuators to replicate the functions of natural muscles (such as strain output, force generation, and power density). Compared with conventional actuators (such as DC motors), these soft actuators tend to be more flexible, conformable, and safer when interacting with humans [2].

A variety of soft actuators available in different materials and functions have been designed to imitate bionic motions. These soft actuators include ionic polymer–metal composite (IPMC) [3,4,5,6], shape memory alloy (SMA) [7,8,9,10], dielectric elastomer actuator (DEA) [11,12,13,14], and twisted and coiled polymer (TCP) [15,16,17]. The IPMC is a popular choice for its low actuation voltage (<5 V) and high flexibility [18]. However, they have limitations of low power density (0.02 W/g) and stress (up to 0.3 MPa) [6]. The unique merits of SMA such as high power density (50 W/g) were also explored [19]. However, SMA actuators suffer from a severe hysteresis effect during actuation. The DEA shosw high power density, but large voltage (10–100 MV/m) is required for actuation [19]. At the same time, the electrodes of the DEA are expensive and hard to produce.

Among these soft actuators, the recently discovered novel TCP is a promising soft actuator candidate for applications in the field of soft robotics [20]. The TCP is fabricated by constant twisting and coiling of polymer fibers or threads. This novel actuator can deliver large strain (over 49%) and high work density (5.3 W/g) via heating [20]. Other advantages such as long durability (over 1 million life cycles) and low cost make it a competitive actuator [21]. Recently, the TCP actuators have been used in applications such as robotic fingers, hands, and arms [22,23,24]. The TCP actuators can be used as an actuating module of soft robots. However, pre-tension and contraction room are necessary for TCP actuation. Tang et al. proposed a multifunctional soft robot module driven by TCP actuators. The contraction room is offered by a bar-like soft body made from silicone [25]. A soft gripper was fabricated to test the performance of the module. In Tang et al.’s research, the cooling speed of TCP actuators is relatively slow due to the closed chamber design. It lead to a long heat-dissipation time after TCP heating. A similar soft robot module was presented by Yang et al., but in their design, TCP actuators were embedded in an open chamber made of rubber to improve the cooling speed [2]. The recovery force was offered by another parallelly placed TCP actuator during the cooling stage. This antagonistic structure reduced the contraction stoke and force of the module significantly.

In this paper, an artificial musculoskeletal actuation module driven by TCP actuators is presented. This module has leaf springs to provide pre-tension for TCP actuators and can be employed as a general unit for robot fabrication. TCP actuators are embedded in the center of the module with two or four leaf springs placed around the TCP. The proposed module is flexible and inexpensive. This module can be potentially applied in aerospace fields (such as space robotics, active flaps, and deployable space structures). This paper is organized as follows. In Section 2, the bio-inspiration and design process of the artificial musculoskeletal actuation module are discussed. In Section 3, the TCP actuators and leaf springs are modeled by considering the displacement and force to improve the actuation efficiency of the module. In Section 4, the module is fabricated by integrating the designed leaf springs and TCP actuators. The performance of the module is demonstrated by experiments. Finally, conclusions and future work are presented.

## 2. Artificial Muscle Unit Design

### 2.1. Bio-Inspiration for the Actuation Module

The human arm accomplishes bending and unbending with the assistance of skeletal muscle. This muscle can be considered as an active element of actuation for contraction or elongation in one direction. These movements are completed through antagonistical muscle structure arrangement and energy from adenosine triphosphate (ATP). When contraction occurs, the muscle will shorten and produce tension, as shown in Figure 1.

Inspired by the structure and contraction model of the skeletal muscle, a kind of bio-mimic musculoskeletal actuation module is proposed. In this module, the contraction motion and force are provided by the TCP muscles, whereas the muscle deformation and recovery force are offered by the leaf springs.

### 2.2. Musculoskeletal Actuation Module Design

The structure of the proposed musculoskeletal actuation module is shown in Figure 2. The main body of the actuation module consists of TCP muscles and leaf springs. These parts are connected by the connection part. The musculoskeletal actuation module is actuated by the contraction of TCP muscles. For type 1 actuation, two leaf springs are uniformly distributed around two TCP muscles with a phase shift of 180° to provide deformation and pre-tension for TCP muscles; For type 2 actuation, four leaf springs are uniformly distributed around the TCP muscles with a phase shift of 90°. Two 3D-printed connection parts are employed at the head and tail of the module to fix and connect the remaining parts.

The driving force of the actuation module is offered by the contraction motion of TCP muscles. When voltage is applied to the TCP muscles, the temperature of the TCP actuators will rise with a contraction along the axial direction. When the voltage is removed, four deformed leaf springs can provide a quick recovery to the initial state for the actuation module.

## 3. Model Formulation

### 3.1. Modelling TCP Muscle

In our previous work [26], the manufacturing process of TCP muscles was treated as the buckling process of an elastic rod. Based on the elastic rod theory, a model was proposed to describe the quantitative relationship between stroke and fabrication parameters (such as fabrication load). The critical load and twist number of coils formed were identified as key factors. According to our model, the stroke of TCP muscles can be obtained when fiber characteristics and fabrication parameters are determined.

According to our previous work [26], the stroke is expressed as:(1)$$\frac{\Delta L}{L}=-\frac{8\left(1+\mu \right)l_{c}\sqrt{F}\alpha_{t}\Delta T^{\prime }}{\pi^{3/2}E^{1/2}d^{2}NL}$$
where *E* is Young’s modulus, μ is Poisson’s ratio, *L* is the unloaded coil length, *N* is the number of coil turns, $l_{c}$ represents fiber length in the coil, *d* represents fiber diameter, *F* is fabrication load, $\alpha_{t}$ denotes coefficient of thermal expansion, and $\Delta T^{\prime }$ is change of temperature.

### 3.2. Modelling Leaf Spring

The leaf springs can provide stable resilience and pre-tension for the actuation module. The leaf springs show radial expansion under linear compression. The linear motion range can be improved by the radial expansion compared to common linear springs.

Chen et al. [27]. proposed that the integrated leaf springs can be approximated by a number of serial-connected small segments (6-DOF linkage). Based on the kinetostatics of the equivalent linkages [28], the force–deflection behavior of the leaf springs can be derived. The configuration of the tip frame of the constructed mechanism $g_{lt}$ and the balance of the elastic deformation can be expressed as [27]
(2)$$\begin{array}{c} \left\{\begin{array}{c} g_{lt}=\prod_{i=1}^{2n}exp\left(\hat{\xi_{i}}\theta_{i}\right)g_{lt,0}\\ \tau =K_{\theta }\theta -J_{\theta }^{T}F\to 0 \end{array}\right. \end{array}$$
where $\theta_{i}$ is the joint variables of the approximated spring, $\hat{\xi_{i}}$ represent the joint twists, $g_{lt,0}$ is the spring’s initial pose in the local frame, 2n is the number of the joints, $K_{\theta }=diag\left(1/c_{1,1},1/c_{1,2},1/c_{2,1},1/c_{2,2},\cdot \cdot \cdot ,1/c_{n,1},1/c_{n,2}\right)$ corresponds to the overall joint stiffness matrix, $J_{\theta }$ denotes the Jacobian matrix, and *F* is the external wrench applied to the tip of the leaf spring.

Thus the force–deflection analysis can be finished by an optimization model, which is represented as
(3)$$\begin{array}{c} min\;c\left(x\right)=\left[\begin{array}{c} \log{\left(g_{t}^{-1}g_{lt}\left(\theta \right)\right)}^{\vee }\\ K_{\theta }\theta -J_{\theta }^{T}F \end{array}\right] \end{array}$$
where $x=\left[\theta ,F\right]\in R^{(2n+7)\times 1}$ relate to the variables of the optimization problem, and $g_{t}\in SE\left(3\right)$ is the target pose for the tip-frame of each leaf spring.

The gradient of the objective function can then be expressed as
(4)$$\nabla =\left[\begin{array}{c} \frac{\partial c}{\partial \theta },\frac{\partial c}{\partial F_{2}} \end{array}\right]=\left[\begin{array}{cc} J_{\theta }, & 0\\ K_{\theta }+K_{J}, & -J_{\theta }^{T} \end{array}\right]$$
where $K_{J}\in R^{2n\times 2n}$ is a configuration-dependent stiffness item. Thus, the update theme for the variables in this equilibrium problem can be written as
(5)$$\begin{array}{c} x^{j+1}=x^{j}+\nabla^{-1}+c^{j} \end{array}$$

The variables will converge stably when $c^{j}$ approaches zero. Thus, the deformation and the recovery force of the leaf spring under linear compression can be obtained. Figure 3 shows the deformation of the leaf spring under different compression levels.

The design of the actuation should meet two design requirements (length matching and force balance). (1) The TCP muscles and the leaf springs have the same axial length; (2) The recovery force offered by the leaf spring is equal to the contraction force provided by TCP muscles in the balance stage. The presented theory can be applied to meet these design requirements. For example, the length requirements can be satisfied by solving Equations (Equation 1) and (Equation 2).

## 4. Results

### 4.1. Fabrication Process

The proposed actuation module consists of TCP muscles, two/four leaf springs, and two 3D-printed connection parts. Silver-coated nylon fiber (PN: 40024104600, Shieldex^®^, Newark, NY, USA) with a diameter of 0.66 mm was used to produce TCP muscles. The fiber is twisted continuously by a DC motor and then annealed to relieve residual stress. Convenient heating and actuation of the TCP muscles can be achieved by the conductive fiber. Two 3D-printed connection parts are used to fix the leaf springs and TCP muscles for module assembly. These connection parts also provide room for electric wires of TCP muscles to connect the power supply. The leaf springs are made from 65 Mn steel alloy. The initial dimensions of the leaf springs are 10 × 190 × 0.15 mm (Type 1) and 8.43 × 190 × 0.1 mm (Type 2).

Two kinds of module configurations are shown in Figure 4. The type 2 (Figure 4b) module has more leaf springs to provide a larger recovery force with quicker recovery and higher response frequency. However, it will suffer from limited working space for the large space deformation of the module. Symmetrical arrangement of the module is necessary because of the deformation characteristic of the leaf spring. It can help to neutralize the negative impact of radial force.

### 4.2. Basic Performance Test

#### 4.2.1. The Control System

The control system was established to heat, measure, and control the TCP muscles and the actuation module. As noted in Section 4.1, silver-coated nylon fiber was employed to fabricate TCP muscles. These muscles can be conveniently heated by electrical power. A microcontroller (AT89C51, Microchip Technology©, Chandler, AZ, USA) was used to compute and process signals. A laser displacement meter (HG-C1100, Panasonic©, Osaka, Japan) was applied to measure the displacement. A micro force sensor (JLBS-M2, Jnsensor©, Bengbu, China) was employed to measure the contraction force. A DAQ module (USB-4716-AE, Advantech©, Shanghai, China) was used to collect and convert signals. These signals were then sent to PC and displayed in GUI for data analysis.

#### 4.2.2. Single TCP Strain and Recovery Stress Test

Performance tests were undertaken to explore the trend of the TCP stroke and force under different temperatures. In these tests, the TCP muscle was actuated under temperatures from 20 °C to 120 °C. The DMA Q800 (TA Instruments, New Castle, DE, USA) was working in tension mode with a frequency of 1 Hz to carry out these tests. The relationship between the temperature and the TCP strain is shown in Figure 5. For given material and fabrication parameters, the TCP strain changes with the temperature (as shown in Equation (Equation 1) and Figure 5). The negative sign in Equation (Equation 1) means that the length of TCP muscles will be shortened when heated. Overall, the predicted results are in agreement with experimental data. The TCP muscle reached the maximal strain of around 25% at 120 °C under an external load of 1.3 N. However, the cooling curve is not strictly fitted with the heating curve (Figure 5). This may be attributed to the lack of recover force at low temperature stage. The initial contraction speed of the TCP muscle at low temperature stage is relatively slow. This can be mainly attributed to the rapid decrease of Young’s modulus and elastic recovery stress at low temperature stage [26]. At higher temperature (above glass transition temperature), the compression force of TCP muscles increases with the amorphous tie molecules modulus, and the inner separation distance is shortened. At the same time, the crystalline bridges are insensitive to the modulus. This brings about the rapid contraction procedure of TCP muscles at higher temperature.

The same experimental setup was arranged with the TCP muscle to measure the contraction force during the actuation stage. The TCP muscle was also actuated under temperatures from 20 °C to 120 °C. Two ends of the TCP muscle were fixed to keep the strain constant (at 30%). The maximal contraction stress of the TCP muscle is approximately 4.3 MPa, as shown in Figure 6.

#### 4.2.3. Musculoskeletal Actuation Module Performance Test

The electrical power for actuation module heating was provided by a DC power supply. Two kinds of musculoskeletal actuation modules were designed, as shown in Figure 4. The deformation process of these modules under applied electrical power can be seen in Figure 7c,d.

The actuation module was actuated in a temperature range from 25 °C to 120 °C. In the actuation module, the contraction force is offered by the thermally driven TCP muscles. This contraction force can be adjusted by actuation temperature, as shown in Figure 7b. The leaf springs can provide recovery force to improve the response frequency of the actuation module. When exposed to electrical power, the contraction force will increase with the actuation temperature in a certain temperature range, and the leaf springs will be compressed. In the recovery stage, the heating voltage will be turned off, and the recovery force provided by the leaf springs will accelerate the recovery procedure of the actuation module. Moreover, feaf springs offer predictable recovery force for the actuation module, as noted in Section 3.2. Contraction room and pre-tension needed for TCP muscle actuation can also be provided by the leaf springs. Irreversible deformation of the TCP muscle will occur at excessively high temperatures. The actuation module reached the maximal contraction force of approximately 0.7 N at 120 °C.

The displacement of the actuation will also increase with the actuation temperature, as shown in Figure 7a. The TCP muscles are elongated by the leaf springs for further contraction when the temperature is low. The actuation module reached the maximal displacement of approximately 6.8 mm at 120 °C. The heating–cooling process of the actuation modules is shown in Figure 7c,d. The modules are heated from 25 °C to 120 °C, then cooled from 120 °C to 25 °C. The contraction speed of the TCP muscles can be improved by employing a higher voltage. The response frequency can also be improved by increasing the heat dissipation area and reducing the cooling period.

## 5. Conclusions

In this paper, we presented a novel kind of artificial musculoskeletal actuation module driven by TCP actuators. A design of the TCP driven module with antagonistic configuration was developed. The design of the actuation module is inspired by skeletal muscle. When actuated, the TCP actuators could deliver contraction motion and force similar to muscle fibers. Leaf springs are employed in this module to provide pre-tension for TCP actuators. They can also improve the recovery speed of the TCP muscles. Thus, the response frequency of the module can also benefit from these structures. The module can contract with a reasonable force and can be used as a general unit for soft robots. The model of the TCP actuators and leaf springs were presented. The established model can be applied for the designing of the module. The capability of the actuation module was tested. Experimental results showed that the actuation module reached the maximal contraction force and displacement of approximately 0.7 N and 6.8 mm at 120 °C under electrical power of 24 V, respectively. The lifespan of the artificial muscle unit depends on the durability of the TCP muscles. According to [20], TCP muscles can deliver over 1 million cycles at an actuation frequency of 1 Hz.

The design of recovery force is significant for improving the response frequency of the actuation module. A silicone shell and traditional springs have been applied to provide recovery force [29,30,31]. However, the driving force of the actuation module may suffer from excessive recovery force and affect the module performance. The leaf springs used in this paper are convenient for recovery force and deformation design. This module showed low-cost, stable repeatability, and ease of fabrication and actuation without the requirements of redundant accessories or special conditions (high voltage). However, there are still some challenges for future application. An open structure was used in this module to improve the passive heat dissipation capacity of the TCP actuators. However, the cooling speed is still relatively slow for a task that needs a high response frequency. More efficient thermal management is needed to improve in this area. In addition, the proposed module can only contract in a straight line. Other motions (such as bending and extension) can be accomplished by future structure design to improve the flexibility.

## Figures and Tables

**Figure 1 materials-15-08261-f001:**
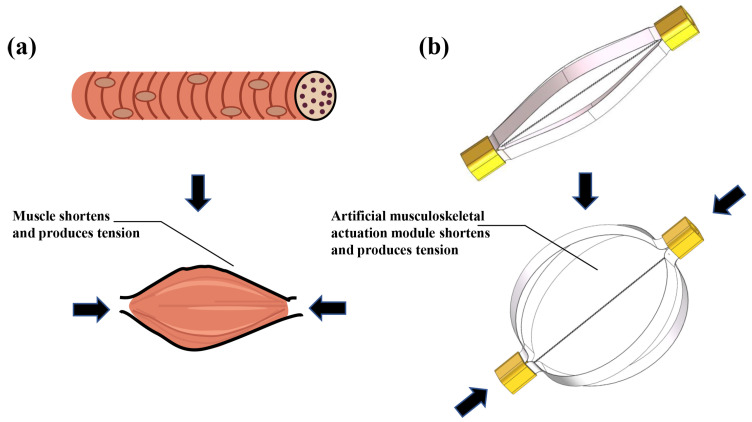
Structure of a bio-inspired actuation module: (**a**) contraction of a muscle fiber; (**b**) contraction of the proposed musculoskeletal actuation module.

**Figure 2 materials-15-08261-f002:**
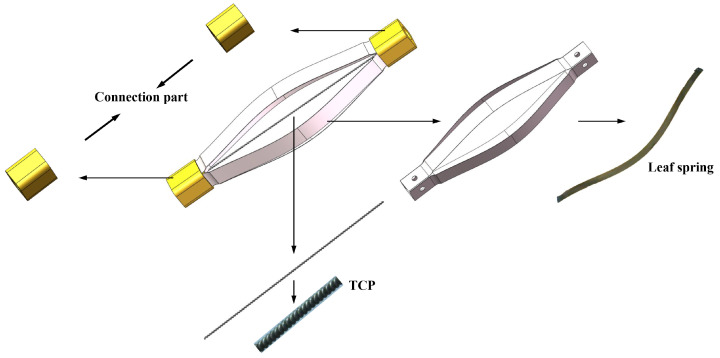
Structure of the proposed musculoskeletal actuation module.

**Figure 3 materials-15-08261-f003:**
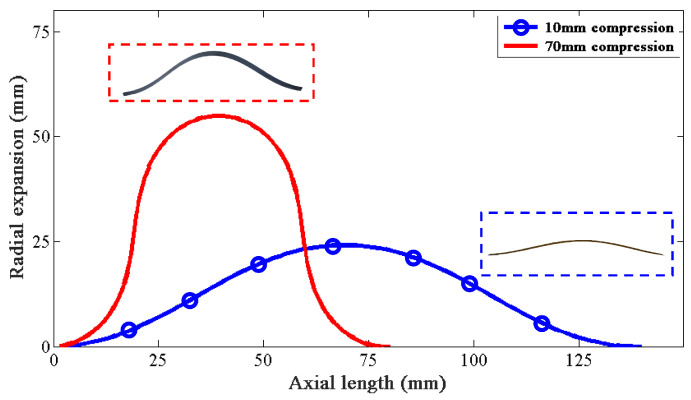
Deformation of a single leaf spring under compression.

**Figure 4 materials-15-08261-f004:**
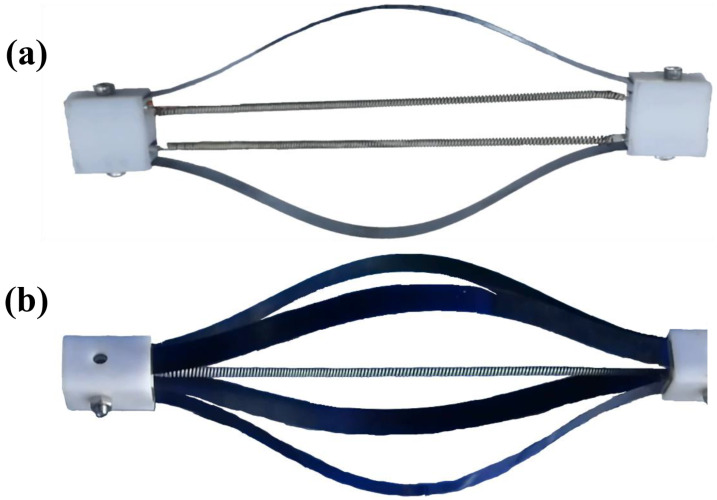
Two kinds of actuation module configurations: (**a**) type 1 actuation module with two leaf springs; (**b**) type 2 actuation module with four leaf springs.

**Figure 5 materials-15-08261-f005:**
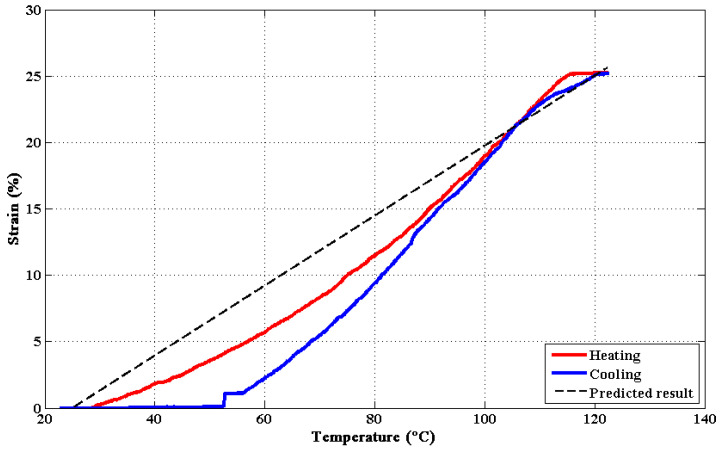
Strain–temperature relationship of the TCP muscles under a load of 1.3 N.

**Figure 6 materials-15-08261-f006:**
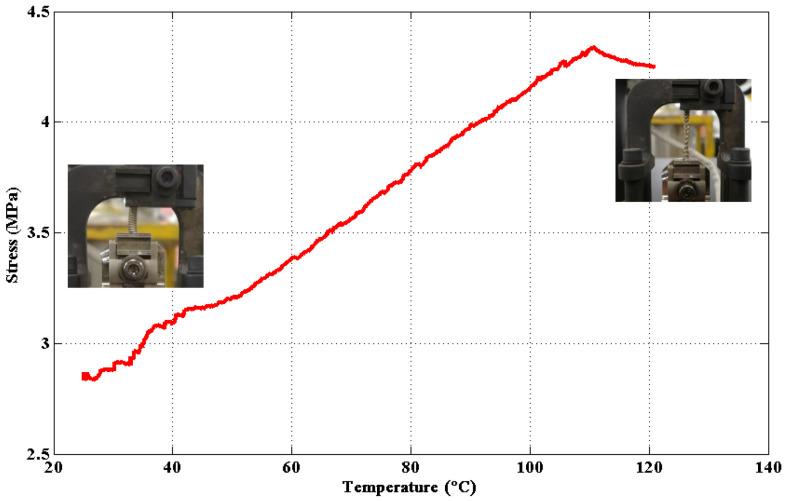
Recovery stress of the TCP muscles from 20 °C to 120 °C.

**Figure 7 materials-15-08261-f007:**
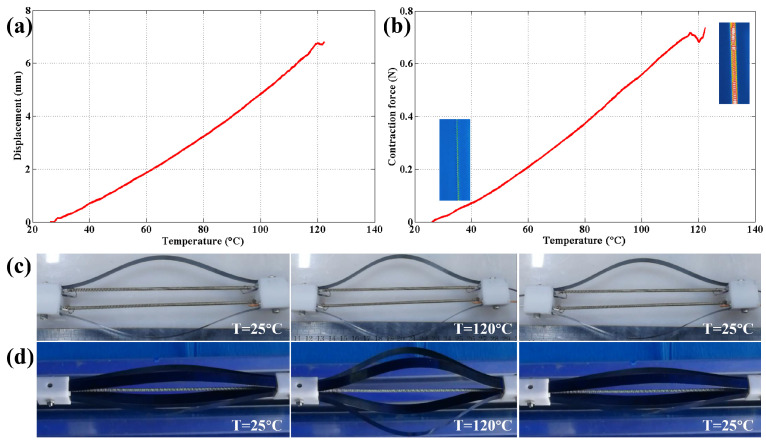
Performance of the musculoskeletal actuation module: (**a**) displacement–temperature curve of type 1 musculoskeletal actuation module; (**b**) force–temperature curve of the 1 musculoskeletal actuation module; (**c**) deformation process of type 1 actuation module; (**d**) deformation process of type 2 actuation module.

## Data Availability

Not applicable.

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
