# Peer review of "Design and Application of a Twisted and Coiled Polymer Driven Artificial Musculoskeletal Actuation Module"

_materials, 2022, doi:10.3390/ma15228261_

Round 1

Reviewer 1 Report

The research presented shows interesting elements, but description of experimental tests and results should be improved.

In general, prototypes should be described better, including more data. As an example: the dimensions of a single spring under compression of figure 3 should be indicated. Type 1 and type 2 of musculoskeletal actuation module should be better presented (dimensions of springs, initial length, etc.).

In the figure 6 there are some incoherence between stress (MPa) and force (N). Please clarify.

In paragraph 4.2.3 it is not clear to witch prototype refer experimental diagrams presented. Please clarify.

Reviewer 2 Report

The very idea of the article is not without sense, but its implementation leaves much to be desired:

1. Abstract is too general. It should be supplemented with concrete results and the most important conclusions.

2. Linguistic proofreading is highly recommended.

3. Eq. 1 is given without explanation and/or literature support.

4. Fig. 5 - in my opinion experiment results are not in accordance with theory, contrary to what Authors wrote.

5. There is no discussion of obtained results with literature, no comparison with other theoretical models and/or realisations.

6. There is almost no connection between presented theory and obtained results, especially the behaviour of the prototype under compression is not tested. Only two experiments showing influence of temperature is obviously not enough.

Reviewer 3 Report

In the manuscript there is presented a novel kind of artificial musculoskeletal actuation module driven by  twisted and coiled polymer (TCP) actuator. Leaf springs are employed in this module to provide pre-tension for TCP actuator. The topic is highly interesting, but unfortunately, there is no presented research on materials as is also stated in the last paragraph of Introduction (In section 2, the bio-inspiration and design process of the artificial musculoskeletal actuation module were discussed. In section 3, the TCP actuators and leaf springs were modeled by considering the displacement and force to improve the actuation efficiency of the module. In section 4, the module was fabricated by integrating the designed leaf springs and TCP actuators. The performance of the module was demonstrated by experiments.).

Further Comments, Notes and Questions:

1. Page 3, Line 74: Please correct “drving force”.

2. Page 3: Eq. (1) is from literature [29]. It is O.K. But has been Eq. (2) published in [30] or [31]? Or did you derive it?

3. Page 5, Fig. 4: There is presented also actuation module with two leaf springs and two TCP muscles, but in Section 2.2 there is described only actuation module with four leaf springs and one TCP muscle. Please extend Section 2.2.

4. Page 5, Line 145: In my opinion better to use “…is shown…”.

5. Page 6 and 7: Missing spaces between values and units “mm”, “N”, “MPa”.

6. Page 6, Fig. 6: Title of the figure is “Contraction force…”, but in the graph there is “Stress” in MPa.

7. Page 6, Lines 159-161: This information has been written already on Page 5 (lines 136-137).

8. Page 7, Line 173: Where it was exactly mentioned in Section 3.2?

9. Page 7, Fig. 7b: Is it really “Recovery force”?

10. Page 7, Fig. 7c: The deformation seems to be very small. Is it true?

11. Page 7, Fig. 7d: Is picture on the right after cooling?

12. Conclusion: It would fine to also mention the achieved response frequency.

13. Conclusion, Line 191-192: “The established model can be applied for the designing of the module”. Have you not applied it so far?

Reviewer 4 Report

The article deals with the issue of artificial muscles, which represent a prospective form of actuators applicable in robotic applications. It is a biologically inspired form of actuators to imitate bionic motions. Their application brings a number of problems that need to be solved. So this article is interesting for readers.
This paper presents a controlled artificial musculoskeletal actuation module that can be used in soft robots.
The article presents a literary overview of similar solved works. Artificial Muscle Unit Design is presented and described.
Model Formulation of the proposed musculoskeletal actuation module is described. Implementation and experimental results are presented. Technically and graphically, the article is prepared at an excellent level, but some important information is missing here. The article needs to be supplemented.

Comments:
Line 74: "The driving force" should be properly "The driving force".
There should be a space between the values ​​of the quantity and the units of the quantity.
The graphs shown in Figures 3, 5, 6, 7 have different text sizes. Please, it needs to be unified.
I did not find the basic parameters of the Artificial Muscle Unit (dimensions, weight, etc.) in the article. With which Artificial Muscle Units the experiments were carried out. What is the effect of dimensions on the behavior of the Artificial Muscle Unit.
For practical purposes, it is also necessary to know the dynamic behavior of the Artificial Muscle Unit in time. Please present the dynamic properties of the Artificial Muscle Unit.
For practical use, it is also interesting to mention the energy requirements of this Artificial Muscle Unit. Also efficiency is quite important and other guidelines for applicability. Another important factor is the lifespan of the Artificial Muscle Unit.

Round 2

Reviewer 1 Report

The article presents in correct way an experimental research on twisted and coiled polymer artificial actuation module. I suggest future work to test dynamic behavior.

Author Response

Thank you for your kind suggestion.  We would like to thank the referee again for taking the time to review our manuscript.

Reviewer 2 Report

The revised manuscript contains corrections significantly improving its quality and predisposing for possible publication.

Hovewer, Authors should consider following remarks:

1. The introduction should indicate the applications related to the subject of the special issue ("Advanced Composite Material Design and Manufacturing Technology for Aerospace Engineering").

2. As Authors wrote in their reply,  "other motion (such as bending and extension) tests will be done in future works", and so the direction of future works, as well as justification of publication of partial results, should be included in the paper. In this context, I also wonder if some indication of this partiality would also be useful in the title/abstract.

Reviewer 3 Report

Dear Authors,

The manuscript can be accepted for publication.

Reviewer

Author Response

We would like to thank the referee again for taking the time to review our manuscript.